# Co-Evolving Agents: Learning from Failures as Hard Negative

## Abstract

The rapid progress of large foundation models has accelerated the development of task-specialized agents across diverse domains. However, the effectiveness of agents remains tightly coupled with the quality of training data, while curating task-specific datasets remains costly and often infeasible in real-world scenarios. Recent work has explored self-improving agents that autonomously generate, refine, and re-train on their own trajectories. A prominent line of approaches further leverages preference optimization by pairing predicted trajectories with scarce ground-truth trajectories, enabling agents to learn directly from their own failures. While these methods outperform supervised fine-tuning, their heavy reliance on predicted trajectories under limited ground-truth supervision leaves them prone to overfitting. To address this, we propose a *co-evolving agents framework* in which a target agent improves jointly with an auxiliary failure agent. The failure agent learns through preference optimization over failure trajectories from both the target and itself, thereby generating hard negatives that are close to success yet remain failures. Incorporating these informative hard negatives into the target agent's optimization sharpens decision boundaries and enhances generalization. Our comprehensive analysis and experiments across benchmark datasets show that our method not only show improved performance but also highlights that failures, instead of being used as-is, can be systematically transformed into structured and valuable learning signals in self-improving agents.

## 1 Introduction

The rapid progress of large foundation models (OpenAI, 2025; Yang et al., 2025; MetaAI, 2025; Anthropic, 2025; Gemini Team, 2025) has facilitated the rise of task-specialized agents across diverse domains, ranging from open-domain dialogue to scientific reasoning tasks (SU et al., 2025; Zeng et al., 2024a; Fu et al., 2025; Bousmalis et al., 2024). These agents inherit the broad generalization capacity of pretrained models, allowing effective adaptation to new tasks with relatively limited supervision. This promise has motivated growing interest in developing methods that adapt foundation models into reliable and effective domain-specialized agents. Recent advances in multi-agent systems and preference optimization further highlight the potential of combining broad pretraining with specialized adaptation. In particular, the ability to automatically curate training signals from agent interactions opens opportunities for scaling beyond static, human-labeled corpora. At the same time, the increasing deployment of agents in dynamic, real-world settings emphasizes the importance of approaches that can continuously refine behavior without costly retraining. Such developments make it timely to revisit how failures, long viewed as undesirable artifacts, can instead be leveraged as constructive learning signals.

Nevertheless, the effectiveness of such agents remains constrained by the quality of task-specific training data (Zhou et al., 2024; Zhao et al., 2024). High-quality datasets are essential for reliable reasoning and decision making, providing the signals required for adaptation to specialized domains. Yet, constructing such datasets is expensive and labor-intensive, often requiring domain expertise and extensive annotation. In many real-world scenarios, constructing large curated datasets is infeasible. In addition to the prohibitive cost of collecting interactions at scale, the need to repeatedly curate data to keep pace with non-stationary environments makes this approach impractical. This bottleneck has motivated growing interest in methods that enable agents to improve autonomously without relying on continuous manual curation (Yuan et al., 2025b; Nguyen et al., 2025; Yin et al.,

Figure 1: Overview of our co-evolving agents framework. The failure agent learns from pairs of failure trajectories and produces *hard negatives*, i.e., high-reward failures that lie close to success. These hard negatives are then incorporated into the target agent's preference optimization, providing more informative contrastive signals. Through this mutual interaction, the two agents co-evolve, enabling the target agent to learn sharper decision boundaries and achieve stronger generalization.

2025). Such methods aim to bridge the gap between the flexibility of foundation models and the rigorous demands of domain-specific tasks, while keeping human intervention to a minimum. A central challenge is therefore to design mechanisms that transform abundant but noisy interaction data into structured supervision that drives reliable improvement.

Self-improving agents (SU et al., 2025; Zeng et al., 2024a; Fu et al., 2025) have emerged as a promising paradigm to reduce reliance on costly human annotation. Maintaining agents at state-of-the-art performance would require continuous human annotation, which is prohibitively costly and infeasible at scale. Instead, self-improving agents automate parts of the data construction process by synthesizing expert-like trajectories from external resources such as documentation or databases, and by repurposing predicted failures as preference data for training. However, in challenging down-stream tasks where pretrained LLMs perform poorly, generating high-quality trajectories themselves remains a major bottleneck, making fully autonomous self-improvement difficult in practice.

In this context, Exploration-Based Trajectory Optimization (ETO) (Song et al., 2024) constructs preference datasets by pairing failure trajectories predicted by the agent with ground-truth ones using a given reward model, enabling preference optimization (Rafailov et al., 2023). Despite the promise of autonomous improvement, this approach remains limited, as it relies on the model's own failure trajectories. These failures are substantially less informative than human-curated negatives, especially in tasks where pretrained LLMs lack prior knowledge. As a result, the dispreferred trajectories offer only weak contrast, making DPO focus on simply increasing the likelihood of expert trajectories rather than shaping a fine-grained decision boundary. This weak supervision often yields limited corrective signals, causing the model to gravitate toward the small expert set and ultimately overfit, rather than acquiring a robust understanding of the underlying task landscape.

To overcome these limitations, we introduce a *co-evolving agents framework* in which a target agent improves jointly with an auxiliary failure agent. The failure agent specializes in preference optimization over failure trajectories from both the target and itself, enabling it to learn a fine-grained landscape of failures rather than merely preferring expert trajectories. Crucially, this design enables the agent to autonomously generate informative *hard negatives* (Robinson et al., 2021; Rafailov et al., 2023; Chen et al., 2020) that are high-reward failures close to success, without any external supervision. Incorporating these hard negatives into the target agent's preference optimization provides stronger and more diverse contrastive signals, sharpening decision boundaries and yielding more generalizable performance.

We validate our framework through comprehensive analysis and experiments across diverse domains, including the online shopping environment WebShop (Yao et al., 2022), the science reasoning environment ScienceWorld (Wang et al., 2022), and the interactive SQL environment Inter-CodeSQL (Yang et al., 2023). Our analysis on these benchmarks verifies that the failure agent does

not simply imitate expert trajectories but continues to generate high-reward failures that serve as informative hard negatives.

Experiments further demonstrate substantial improvements over competitive baselines, achieving large margins of gain across benchmarks and reflecting stronger generalization to diverse tasks. These findings highlight that systematically harnessing failures as structured learning signals, rather than treating them as byproducts, opens a promising direction for advancing self-improving agents.

Our contributions are summarized as follows:

- We introduce a *failure agent* that, unlike prior frozen negative agents trained on human-curated data, continuously learns from failure trajectories and captures a fine-grained failure landscape.

- We propose a *co-evolving agents framework* where a target and failure agent improve jointly, with the failure agent generating hard negatives that sharpen decision boundaries and enhance generalization.

- Our experiments further confirm that failures, when systematically harnessed, can be transformed into structured learning signals that drive more robust self-improving agents.

## 2 RELATED WORK

**Self-Improving Agents** Building high-performing agents requires high-quality datasets, which are costly and often infeasible in real-world scenarios. Self-improving agents address this by autonomously generating, refining, and reusing data for continual learning. Some approaches synthesize trajectories from tutorials, documentation, or persona hubs (SU et al., 2025; Zeng et al., 2024a; Fu et al., 2025), while others use planning methods such as Monte Carlo Tree Search (MCTS) (Yuan et al., 2025b). Beyond dialogue, self-improvement has been explored in programmatic action composition (Nguyen et al., 2025), robotics (Bousmalis et al., 2024), and code generation (Yin et al., 2025). Another line leverages failure trajectories paired with expert ones for preference optimization (Song et al., 2024; Xiong et al., 2024), but these typically use failures as-is, limiting generalization. Multi-agent variants (Zhang et al., 2024) employ negative agents trained on curated failure datasets, yet these are frozen and restricted to dialogue tasks, offering only limited benefit compared to success-based supervision.

**Hard Negatives in Contrastive Optimization** Reinforcement Learning from Human Feedback (RLHF) (Lee et al., 2024) has been the standard paradigm for aligning language models, but it requires costly reward modeling and policy optimization. Recent contrastive methods such as Direct Preference Optimization (DPO) (Rafailov et al., 2023) and Generalized Preference Optimization (GRPO) (Tang et al., 2024) simplify this process by directly optimizing policies on preference pairs, bypassing explicit reward models. At the core of contrastive optimization is the idea that learning benefits most from informative comparisons. In particular, *hard negatives* that are difficult to distinguish from the preferred ones and thus yield small preference margins, are known to provide stronger supervision and promote sharper decision boundaries (Robinson et al., 2021; Rafailov et al., 2023; Chen et al., 2020).

## 3 PRELIMINARIES

The interaction between an LLM agent and its environment can be formalized as a partially observable Markov decision process (POMDP) () $(\mathcal{U}, \mathcal{S}, \mathcal{A}, \mathcal{O}, T, R)$, as in Song et al. (2024). Here, $\mathcal{U}$ denotes the instruction space, $\mathcal{S}$ the state space, $\mathcal{A}$ the action space, $\mathcal{O}$ the observation space, $T : \mathcal{S} \times \mathcal{A} \to \mathcal{S}$ the transition function, and $R : \mathcal{S} \times \mathcal{A} \to [0, 1]$ the reward function. In our LLM-agents setting, $\mathcal{U}, \mathcal{A}$, and $\mathcal{O}$ are expressed in natural language.

At the beginning of each episode, the agent receives an instruction $u \in \mathcal{U}$ and generates its first action $a_1 \sim \pi_\theta(\cdot \mid u) \in \mathcal{A}$ from its policy $\pi_\theta$ parameterized by $\theta$. The action updates the latent state $s_t \in \mathcal{S}$ and produces an observation $o_t \in \mathcal{O}$. Subsequent actions are conditioned on the full interaction history, so that

$$a_t \sim \pi_\theta(\cdot \mid u, a_1, o_1, \ldots, a_{t-1}, o_{t-1}) \in \mathcal{A}.$$

This process unfolds until either the task is solved or the step budget is exceeded. A trajectory can therefore be written as

$$e = (u, a_1, o_1, \ldots, o_{n-1}, a_n) \sim \pi_\theta(e \mid u), \tag{1}$$

with likelihood

$$\pi_\theta(e \mid u) = \prod_{j=1}^{n} \pi_\theta(a_j \mid u, a_1, o_1, \ldots, o_{j-1}), \tag{2}$$

where $n$ is the trajectory length.

Finally, a reward $r(u, e) \in [0, 1]$ is assigned to the trajectory, where $r(u, e) = 1$ corresponds to full task success and lower values indicate partial or failed attempts. This formulation sets up the basis for preference-based training methods that compare trajectories according to their rewards.

## 4 METHOD

In this section, we propose our *co-evolving agents framework* in which a target agent and a failure agent are trained in alternating phases and gradually improve through mutual interaction. In Section 4.1, we describe the behavioral cloning stage used to initialize the base policy from supervised fine-tuning on expert trajectories. Next, in Section 4.2, we introduce the failure agent, which learns via preference optimization over failure trajectories from both the target and itself and generates fine-grained hard negatives that are close to success yet still failures. Finally, in Section 4.3, we describe how the target agent leverages expert trajectories, its own predicted failures, and failures generated by the failure agent to construct diverse preference datasets for direct preference optimization (DPO) (Rafailov et al., 2023). By training on this richer set of comparisons and alternating with the failure agent, the target agent achieves more effective learning and stronger generalization within a co-evolutionary loop. The overall pipeline is illustrated in Figure 1.

### 4.1 BEHAVIORAL CLONING WITH SUPERVISED FINE-TUNING

We first initialize a base policy through behavioral cloning, which equips the agent with fundamental task-solving ability before self-improvement. Given an expert dataset $\mathcal{D} = \{(u^{(i)}, e^{(i)})\}_{i=1}^{|\mathcal{D}|}$, each trajectory $e = (u, a_1, o_1, \ldots, a_n)$ consists of a task instruction $u$, actions $a_t \in \mathcal{A}$, and observations $o_t \in \mathcal{O}$. The agent policy $\pi_\theta$ is trained with an autoregressive supervised fine-tuning (SFT) objective:

$$\mathcal{L}_{\text{SFT}}(\theta) = -\mathbb{E}_{e \sim \mathcal{D}} \big[ \log \pi_\theta(e \mid u) \big], \tag{3}$$

where the trajectory likelihood decomposes as

$$\pi_\theta(e \mid u) = \prod_{t=1}^{n} \pi_\theta(a_t \mid u, a_{<t}, o_{<t}). \tag{4}$$

In practice, the instruction, actions, and observations are concatenated into a single text sequence $t = (t_1, t_2, \ldots, t_l)$. The loss is then computed by applying the autoregressive likelihood only to tokens corresponding to agent actions:

$$\mathcal{L}_{\text{SFT}}(\theta) = -\sum_{k=1}^{l} \log \pi_\theta(t_k \mid t_{<k}) \cdot \mathbf{1}(t_k \in \mathcal{A}), \tag{5}$$

where $\mathbf{1}(t_k \in \mathcal{A})$ is an indicator that selects tokens generated as agent actions.

This supervised fine-tuning stage provides the base policy $\pi_{\text{base}}$, which serves as the starting point for co-evolution with the failure agent. To ensure simplicity, both target and failure agents are initialized from independently trained base policies on the same expert dataset, providing a comparable starting point while allowing only minor stochastic differences.

### 4.2 FAILURE AGENT FOR GENERATING HARD NEGATIVES

We introduce an auxiliary *failure agent* $\pi_{\theta_f}$ whose role is to specialize in unsuccessful trajectories and refine them into informative hard negatives (). Unlike the target agent $\pi_{\theta_t}$, which is optimized

toward expert success, the failure agent focuses on modeling the space of failures and extracting fine-grained signals from them. This complementary specialization enables the two agents to co-evolve in alternating phases.

**Preference Dataset.** The preference dataset for the failure agent consists of failure trajectories generated by both the target and itself. Formally, let $\mathcal{F}_{\text{tgt}} = \{e_{\text{tgt}} \mid r(u, e_{\text{tgt}}) < 1\}$ and $\mathcal{F}_{\text{fail}} = \{e_{\text{fail}} \mid r(u, e_{\text{fail}}) < 1\}$ denote the sets of failure trajectories generated by the target and failure agents, respectively. We construct a preference dataset by pairing failures with different reward levels:

$$\mathcal{D}_{\text{fail}} = \Big\{ (u, e_{\text{chosen}}, e_{\text{rejected}}) \,\big|\, e_{\text{chosen}}, e_{\text{rejected}} \in \mathcal{F}_{\text{tgt}} \times \mathcal{F}_{\text{fail}}, \ r(u, e_{\text{chosen}}) > r(u, e_{\text{rejected}}) \Big\}. \quad (6)$$

Here, both $e_{\text{chosen}}$ and $e_{\text{rejected}}$ are failure trajectories, and their relative preference is determined by evaluating rewards with the given reward model, where the higher-reward trajectory is selected as $e_{\text{chosen}}$ and the lower as $e_{\text{rejected}}$. This construction enables the failure agent to exploit not only its own generated failures but also those produced by the target, providing a richer set of pairwise comparisons.

**Preference Optimization.** We adopt the direct preference optimization (DPO) objective (Rafailov et al., 2023) for training on failure trajectories. Given a reference policy $\pi_{\text{ref}}$, the failure agent $\pi_{\theta_f}$ is updated by

$$\mathcal{L}_{\text{DPO}}(\theta_f) = -\mathbb{E}_{(u, e_{\text{chosen}}, e_{\text{rejected}}) \sim \mathcal{D}_{\text{fail}}} \left[ \log \sigma \Big( \beta \log \frac{\pi_{\theta_f}(e_{\text{chosen}} | u)}{\pi_{\text{ref}}(e_{\text{chosen}} | u)} - \beta \log \frac{\pi_{\theta_f}(e_{\text{rejected}} | u)}{\pi_{\text{ref}}(e_{\text{rejected}} | u)} \Big) \right], \quad (7)$$

where $u$ is the task instruction, $\sigma(\cdot)$ is the logistic sigmoid, and $\beta$ is a scaling factor. This objective drives the failure agent to distinguish between relatively better and worse failures, thereby capturing subtle distinctions within the failure space.

**Hard Negatives.** By constructing preference datasets over pairs of failure trajectories and training with the DPO objective, the failure agent learns to capture nuanced differences among failures rather than only aligning with expert trajectories. This enables fine-grained learning of the failure trajectory landscape by considering diverse failure cases and, in particular, generating near-success failures that remain informative despite not solving the task. Such hard negatives provide informative signals that cannot be obtained from simple expert-versus-failure comparisons, allowing the model to establish a sharper decision boundary between success and failure. Please note that we define hardness based on the relative reward value rather than using a fixed threshold. Within the 0–1 reward range where success corresponds to 1.0, a failure trajectory with a reward of 0.8 is considered harder than one with 0.5 because it lies closer to the success reward. Incorporating these refined failure trajectories into the target agent's optimization enables the model to learn sharper decision boundaries and achieve stronger generalization. To better understand the role of the failure agent, we further conduct both quantitative and qualitative analyses of the generated failure trajectories (Section 5.2).

## 4.3 Co-evolutionary Training

The target agent improves by incorporating failure trajectories generated by the failure agent into its preference optimization. We construct a preference dataset $\mathcal{D}_{\text{tgt}}$ consisting of three types of trajectory pairs: (i) expert trajectories versus target-predicted trajectories, (ii) expert trajectories versus failure-agent trajectories, and (iii) failure trajectories from the target versus those from the failure agent. Formally,

$$\mathcal{D}_{\text{tgt}} = \Big\{ (u, e_{\text{chosen}}, e_{\text{rejected}}) \,\big|\, (e_{\text{chosen}}, e_{\text{rejected}}) \in \{(e_{\text{exp}}, e_{\text{tgt}}), (e_{\text{exp}}, e_{\text{fail}})\} \cup (\mathcal{F}_{\text{tgt}} \times \mathcal{F}_{\text{fail}}) \Big\}, \quad (8)$$

where $e_{\text{exp}}$ denotes expert trajectories, $e_{\text{tgt}}$ target-predicted trajectories, and $e_{\text{fail}}$ failure-agent trajectories. Here, $\mathcal{F}_{\text{tgt}} = \{e_{\text{tgt}} \mid r(u, e_{\text{tgt}}) < 1\}$ and $\mathcal{F}_{\text{fail}} = \{e_{\text{fail}} \mid r(u, e_{\text{fail}}) < 1\}$ denote the sets of failed trajectories generated by the target and failure agents, respectively.

The target agent is optimized with a weighted DPO objective (Rafailov et al., 2023) together with an auxiliary supervised fine-tuning (SFT) loss on the chosen trajectories:

$$\mathcal{L}_{\text{target}} = \lambda_{\text{DPO}} \, \mathcal{L}_{\text{DPO}} + \lambda_{\text{SFT}} \, \mathbb{E}_{(u, e_{\text{chosen}}) \sim \mathcal{D}_{\text{tgt}}} \big[ -\log \pi_\theta(e_{\text{chosen}} \mid u) \big]. \quad (9)$$

| Dataset | Train | Test | | Action Space | Max Turns |
|---|---|---|---|---|---|
| | | Seen | Unseen | | |
| WebShop | 1624 | 200 | – | 8 | 10 |
| ScienceWorld | 1483 | 194 | 241 | 16 | 100 |
| InterCodeSQL | 1500 | 200 | – | $\infty$ (SQL) | 10 |

Table 1: Overview of the benchmark statistics. Test-Seen and Test-Unseen indicate test sets constructed from seen and unseen scenarios, respectively. Action Space denotes the number of available actions, and Max Turns specifies the maximum number of interaction turns in the expert trajectories.

As noted by Yuan et al. (2025a), DPO alone maximizes relative preference margins but can become unstable, since the space of chosen trajectories is much smaller than that of rejected ones. This imbalance may lead the model to over-penalize rejected samples while insufficiently reinforcing preferred ones. To stabilize training, we introduce the auxiliary SFT loss on the chosen trajectories, which grounds the policy toward high-reward behaviors. For all datasets, we set $\lambda_{\text{DPO}} = 1.0$ and $\lambda_{\text{SFT}} = 0.1$ as the default weighting.

Additionally, since expert–prediction pairs may arise from both the target agent's and the failure agent's trajectories, we assign per-pair weights such that the total weight of all expert–prediction pairs generated for a given instruction sums to 1.0. This prevents these pairs from dominating the training signal while still encouraging alignment toward high-reward behaviors. For failure–failure pairs, we apply DPO exclusively by setting $\lambda_{\text{SFT}} = 0$, relying purely on preference optimization to avoid introducing misleading supervision from incorrect trajectories.

This design prevents over-counting of expert signals while ensuring that failure pairs receive full emphasis. Failure–failure comparisons are especially valuable, as they capture subtle distinctions between suboptimal behaviors, thereby sharpening the agent's decision boundaries. As training alternates with the failure agent, the two agents form an implicit arms race: the failure agent generates increasingly challenging negatives, while the target agent learns to overcome them. This co-evolutionary loop not only enhances robustness within the training domain but also improves generalization to unseen environments where expert supervision is limited.

## 5 EXPERIMENTS

### 5.1 EXPERIMENTAL SETTINGS

**Datasets** We conduct experiments on three representative benchmarks: *WebShop* for web navigation, *ScienceWorld* for scientific reasoning, and *InterCodeSQL* for interactive SQL querying. All three environments provide continuous final rewards in $[0, 1]$, enabling fine-grained evaluation of task completion. Expert trajectories are collected through a combination of human annotations and GPT-4–assisted generation in the ReAct format (Yao et al., 2023), with additional filtering based on final rewards to ensure quality. We present the overview statistics in Table 1. Also, Example trajectory samples for each dataset and further details of the environments and trajectory collection process are provided in Appendix A.

**Implementation Details** We adopt Llama-2-7B-Chat (Touvron et al., 2023) and Qwen3-4B-Instruct-2507 (Yang et al., 2025). All models are optimized with AdamW (Loshchilov & Hutter, 2017), and each training phase is performed for 3 epochs with co-evolution iterations set to 3 for WebShop and ScienceWorld and 5 for InterCodeSQL. All other hyperparameters are kept identical across datasets to ensure fair comparison. Experiments are conducted on 8 NVIDIA H100 GPUs with 80GB memory, and further implementation details are provided in Appendix B.

**Baselines** We compare our framework with standard imitation learning and several strong post-imitation baselines following Song et al. (2024). Supervised fine-tuning (SFT) (Chen et al., 2023; Zeng et al., 2024b) trains agents via behavioral cloning on expert trajectories and serves as the base policy for other methods. Rejection Fine-Tuning (RFT) (Yuan et al., 2023) augments the expert dataset with success trajectories identified by rejection sampling, while Proximal Policy Optimization (PPO) (Schulman et al., 2017) directly optimizes the SFT policy with reinforcement learning

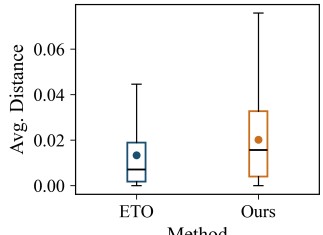 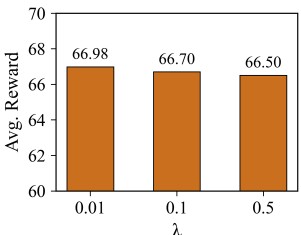

Figure 3: Diversity analysis          Figure 4: Ablation on $\lambda_{\text{SFT}}$.

to maximize task rewards. We additionally include ETO (Song et al., 2024), which applies DPO over expert-versus-predicted pairs without the co-evolution mechanism, thus serving as a DPO-only baseline for isolating the contribution of failure-agent learning. For reference, we also report results from GPT-3.5-Turbo (Ouyang et al., 2022) and GPT-4 (Achiam et al., 2023) with in-context learning. We report average reward as the primary evaluation metric.

## 5.2 ANALYSIS ON FAILURE TRAJECTORIES

To better understand the role of the failure agent, we conduct a quantitative and qualitative analysis of the generated failure trajectories.

### 5.2.1 QUANTITATIVE ANALYSIS

We analyze the failure trajectories generated by ETO and our method along three dimensions: (i) the distribution of successful, negative, and hard-negative trajectories produced during exploration and (ii) the diversity of generated failure trajectories. As in the baselines, we perform a single rollout per instruction due to the high computational cost of long, multi-turn trajectories.

**The distribution of generated trajectories** We separate negative and hard-negative trajectories using a reward threshold of 0.6. Although a higher threshold would be preferable for identifying ideal hard negatives, trajectories with reward above 0.7–0.8 appear in fewer than 1% of cases in current self-improving agents. The resulting statistics are summarized in Table 2. Across all benchmarks, our method generates substantially more negative and hard-negative trajectories than ETO. Specifically, negative trajectories increase by about 9.5% in WebShop, 16.7% in ScienceWorld, and 9.0% in InterCodeSQL. Hard negatives similarly increase by 2.3%, 8.7%, and 4.3%, respectively. These consistent gains indicate that the failure agent effectively expands the informative failure space.

| Task | Method | Success ↓ | Failure ↑ | Hard Neg. ↑ |
|---|---|---|---|---|
| WebShop | ETO | 51.4% | 25.9% | 22.7% |
| | Ours | **39.6%** | **35.4%** | **25.0%** |
| ScienceWorld | ETO | 75.3% | 19.3% | 5.4% |
| | Ours | **49.9%** | **36.0%** | **14.1%** |
| InterCodeSQL | ETO | 58.7% | 37.7% | 3.2% |
| | Ours | **45.8%** | **46.7%** | **7.5%** |

Table 2: Statistics of generated trajectories.

**The diversity of generated failure trajectories** We measure the diversity of failure trajectories by embedding each generated trajectory using the LLM2Vec–Meta-Llama-3-8B-Instruct-mntp BehnamGhader et al. (2024) as a sentence-embedding model and computing the average pairwise distance among trajectories generated per instruction during training. As shown in Figure 3, the failure agent produces a larger number of trajectories with higher distances compared to ETO, and the boxplot demonstrates a notably larger mean and standard deviation. This confirms that the

failure agent explores a broader and more diverse failure space rather than overfitting to specific modes.

Additionally, although the failure agent is trained only on failure trajectories, we observe that it practically does not collapse in our experiments. Its joint training on the target agent's failure trajectories, which often contain partial progress toward the goal, allows it to maintain a comparable level of competence and prevents degeneration into trivial failures.

### 5.2.2 QUALITATIVE ANALYSIS

We qualitatively examine the role of hard negatives through representative examples, focusing on (i) the quality of the hard negatives and (ii) the effect of hard negatives.

**The quality of the hard negatives**   We analyze the WebShop task of buying a machine-washable curtain. As shown in the example below, ETO produces a shallow failure that ignores key constraints such as washability and price. In contrast, our method generates a more structured failure: the agent systematically filters mismatching candidates, checks all relevant attributes, verifies constraints, and arrives at a nearly correct choice. Although still unsuccessful, the trajectory exhibits a coherent decision process and serves as a prototypical hard negative, reflecting substantially higher quality than the failures produced by ETO. Additional examples are provided in Appendix C.

---

**Machine-Washable Curtains (52"×90")**

**Instruction:** I need a machine-washable curtain for the living room, sized 52" wide by 90" long, priced under $60.00.

---

**ETO:** The agent clicks an early search result, selects the 52"×90" option, and buys it without verifying washability, comparing alternatives, or checking that the final price meets the budget.
*Reward:* 0.50    *Steps:* 4    *Outcome:* Failure

---

**Ours:** The agent navigates through multiple product pages, filtering by washability, size, and price. It identifies a curtain with a 52"×90" option, verifies that it is machine-washable and within budget, and chooses the matching size variant before purchasing.
*Reward:* 0.75    *Steps:* 8    *Outcome:* Failure

---

**The effect of hard negatives**   We examine how these trajectories influence the target agent's behavior. As illustrated in the ScienceWorld example of growing a lemon shown below, ETO fails to make meaningful progress: after planting seeds, it repeatedly issues waiting or invalid actions and does not attempt soil preparation, tool use, or environment control. These shallow failures provide limited supervision for preference optimization.

In contrast, our method produces hard-negative failures that attempt multiple sub-skills and carry out most of the required pipeline, including navigation, soil collection, soil preparation, planting, and greenhouse regulation. Although these trajectories still fail to achieve the final goal, they exhibit coherent multi-step behavior that provides substantially richer supervision for DPO. As a result, the target agent acquires these sub-skills more effectively and completes the task more reliably in subsequent iterations. A full version of this example is provided in Appendix C.

---

**Growing a Lemon with Cross-Pollination**

**Instruction:** Grow a lemon by planting seeds, preparing soil, and enabling cross-pollination.

---

**ETO — Prediction:** Plants seeds directly into water-filled pots and then loops through `wait`/`look around` or invalid focus actions. No soil preparation, tool use, or environment control is attempted.
*Reward:* 0.25    *Steps:* 60    *Outcome:* Failure

---

> **ETO — Trained Failure:** Repeats similar shallow behavior and never establishes the pre-conditions needed for growth or pollination.
> *Reward:* 0.25    *Steps:* 49    *Outcome:* Failure
>
> ---
>
> **Ours — Prediction:** Collects soil with a shovel, fills pots, plants seeds, waits for flowering, and regulates the greenhouse by closing doors before focusing on the lemon.
> *Reward:* 1.00    *Steps:* 46    *Outcome:* Success
>
> **Ours — Trained Failure:** Performs the full soil–plant–pollination pipeline with correct subskills but fails at the final focus action, forming a high-quality hard negative.
> *Reward:* 0.50    *Steps:* 60    *Outcome:* Failure

### 5.3 Main Results

We evaluate our framework on three challenging multi-step decision-making benchmarks: WebShop for web navigation, ScienceWorld for scientific reasoning, and InterCodeSQL for interactive SQL querying. All benchmarks use a normalized reward in $[0, 1]$, enabling a fine-grained comparison of task success.

**Results on Llama-2.** Table 3 summarizes the results on Llama-2 across WebShop, Science-World, and InterCodeSQL. Under in-context learning without fine-tuning, GPT-4 and GPT-3.5-Turbo achieve 63.2 and 62.4 on WebShop, but their performance drops sharply on ScienceWorld and InterCodeSQL, illustrating the limitations of prompt-only adaptation for reasoning-intensive or previously unseen domains.

Fine-tuning methods substantially improve performance by aligning models with environment-specific interaction patterns. Among these methods, ETO serves as a strong overall baseline with an average reward of 58.3 across benchmarks. Please note that the results of PPO and RFT on ScienceWorld are taken from Song et al. (2024); Xiong et al. (2024).

Our method achieves an average reward of 64.1, outperforming ETO (58.3) by +5.8%. The improvements are consistent across all benchmarks: WebShop (+3.7%), seen ScienceWorld (+4.1%), unseen ScienceWorld (+6.5%), and InterCodeSQL (+4.4%). The largest gain appears on the unseen ScienceWorld split, indicating that leveraging diverse, near-success hard negatives leads to substantially stronger generalization to unfamiliar scientific environments.

**Results on Qwen3.** Table 4 reports results using Qwen3. Compared to Llama-2, Qwen3 shows a stronger SFT baseline on WebShop (63.9) but exhibits slightly limited performance on Science-World and particularly low performance on InterCodeSQL. This reflects differences in the pretrained models' prior knowledge, including weaker grounding in scientific reasoning and minimal prior exposure to SQL-style interactive querying.

Fine-tuning with ETO provides substantial gains, achieving an average reward of 59.5 across benchmarks. Our method further improves performance to 66.3, surpassing ETO by +6.8%. The improvements are consistent across all tasks: WebShop (+6.8%), seen ScienceWorld (+6.5%), unseen ScienceWorld (+3.3%), and InterCodeSQL (+10.3%). Notably, the gains on both seen and unseen ScienceWorld splits indicate that the proposed failure-agent framework generalizes well even with smaller model scales.

Overall, these results demonstrate that our approach provides consistent and substantial improvements over ETO across architectures, highlighting the robustness of leveraging diverse, near-success hard negatives produced through co-evolution.

**Comparison with DART-style Multi-Sampling.** To strengthen the RFT baseline, we incorporated a DART-style multi-sample setting using a rollout budget of $N = 10$ and $n = 5$ samples to estimate instruction-level difficulty. The difficulty score determines how many additional rollouts to sample, and all resulting successful trajectories are added to the RFT-style SFT training set.

However, the effect remains limited: RFT achieves 60.89 on WebShop, and RFT+DART increases this only to 61.90 on Llama-2. We attribute this to the nature of our benchmarks—unlike math do-

| Adaptation | Models | WebShop | ScienceWorld | | InterCodeSQL | Avg. |
|---|---|---|---|---|---|---|
| | | | Seen | Unseen | | |
| In-context | GPT-4 | 63.2 | 42.9 | 38.1 | 38.5 | 45.7 |
| | GPT-3.5-Turbo | 62.4 | 7.9 | 10.5 | 37.8 | 29.7 |
| Fine-tuning | Llama-2-7B-Chat + SFT | 59.2 | 47.3 | 41.9 | 30.8 | 44.8 |
| | Llama-2-7B-Chat + PPO | 64.2 | 59.4 | 51.7 | 52.4 | 56.9 |
| | Llama-2-7B-Chat + RFT | 61.3 | 71.6 | 54.3 | 35.6 | 55.7 |
| | Llama-2-7B-Chat + ETO | 63.0 | 65.6 | 55.5 | 49.4 | 58.3 |
| | Llama-2-7B-Chat + Ours | **66.7** | **69.7** | **62.0** | **53.8** | **64.1** |

Table 3: Main results.

| Adaptation | Models | WebShop | ScienceWorld | | InterCodeSQL | Avg. |
|---|---|---|---|---|---|---|
| | | | Seen | Unseen | | |
| Fine-tuning | Qwen3-4B-Instruct-2507 + SFT | 63.9 | 43.6 | 40.8 | 12.7 | 40.3 |
| | Qwen3-4B-Instruct-2507 + ETO | 65.7 | 58.6 | 55.2 | 58.8 | 59.5 |
| | Qwen3-4B-Instruct-2507 + Ours | **72.5** | **65.1** | **58.5** | **69.1** | **66.3** |

Table 4: Results on Qwen3-4B-Instruct-2507

mains where pretrained LLMs have strong priors and multi-sampling often yields correct solutions, our tasks require grounded, environment-specific reasoning (web navigation, scientific procedures, SQL), where success via multi-sampling is rare. Consequently, difficulty-based selection offers only modest benefit in this setting.

## 5.4 ABLATION STUDY

**Ablation on Varying $\lambda_{\text{SFT}}$.** We conduct an ablation study to examine the sensitivity of our method to the SFT weight $\lambda_{\text{SFT}}$. As shown in Figure 4, the average reward remains stable across a wide range of values (0.01, 0.1, 0.5), suggesting that the proposed method is reasonably robust as long as the SFT term is kept weaker than DPO.

**Parameter-Matched Ablation on the Failure Agent.** To ensure that the performance gains do not arise from an increased parameter budget, we perform a parameter-matched ablation in which the failure agent is replaced with an auxiliary positive agent trained in the same manner as ETO using only expert and the agent's own failures. This variant keeps the total number of trainable parameters identical to our co-evolving framework. The positive-agent baseline achieves an average reward of 62.8 on WebShop, which is comparable to ETO and clearly below the 66.7 obtained with our method. These results indicate that explicitly modeling and refining failures provides substantial benefit beyond what can be achieved by simply adding another success-oriented agent.

## 6 CONCLUSION

We introduced a co-evolving agents framework where a target agent and a failure agent learn in alternating phases and improve through mutual interaction. By training the failure agent on failure–failure preferences, it generates near-success failures that serve as informative hard negatives. Incorporating these trajectories into the target agent's preference optimization sharpens decision boundaries and improves robustness and generalization. Experiments across WebShop, Science-World, and InterCodeSQL demonstrate consistent gains in diverse domains, underscoring that failures, when refined into structured signals, can be transformed into valuable resources for self-improving agents. We hope our findings facilitate more principled handling of failure trajectories, ultimately contributing to the advancement of the next generation of self-improving agents.

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

# A DATASETS

**WebShop.**  WebShop (Yao et al., 2022) is a goal-oriented web navigation environment where an agent interacts with an e-commerce platform to fulfill natural language shopping instructions. A final reward in $[0, 1]$ is assigned based on how well the purchased item matches the instruction in terms of attributes and price. For training, we use the expert and GPT-4 ReAct-style trajectories provided by Song et al. (2024), retaining only those with final rewards above 0.7 to ensure quality.

---

### Example WebShop Trajectory

**Instruction:**
I need a long clip-in hair extension which is natural looking, and price lower than 40.00 dollars.

**Conversation:**

- **Human:** You are web shopping. I will give you instructions about what to do. Every round I will give you an observation and a list of available actions...
- **GPT:** OK
- **Human:** WebShop [SEP] Instruction: [SEP] i need a long clip-in hair extension ...
- **GPT:** *Thought: I need to find a long clip-in hair extension ...* **Action:** `search[long clip-in hair extension]`
- **Human:** Observation: ... (product list with IDs and prices)
- **GPT:** *Thought: The first option B09C337K8S fits the budget and is natural looking...* **Action:** `click[b09c337k8s]`
- **Human:** Observation: ... (product details shown)
- **GPT:** *Thought: The details confirm it is natural looking and affordable.* **Action:** `click[buy now]`

**Reward:** 1.0

---

**ScienceWorld.**  ScienceWorld (Wang et al., 2022) is a text-based embodied environment for conducting elementary science experiments across ten domains, including thermodynamics and electricity. Each task consists of multiple optional subgoals, and the final reward in $[0, 1]$ reflects the proportion of subgoals completed. Following Song et al. (2024), we exclude the two longest task types and subsample test instances with more than ten variations. We evaluate generalization using the original test set of unseen variations (e.g., training on boiling water and testing on boiling lead) and use the development set for seen scenarios. Expert trajectories are drawn from the official dataset, with additional GPT-4–generated trajectories filtered by final rewards, as in Song et al. (2024).

---

### Example ScienceWorld Trajectory

**Instruction:**
Your task is to determine if a metal pot is electrically conductive. The metal pot is located in the kitchen. If conductive, place it in the blue box; if nonconductive, place it in the orange box.

**Conversation:**

- **Human:** You are a helpful assistant to do some scientific experiment in an environment. (Lists available rooms and actions.)
- **GPT:** OK
- **Human:** Your task is to determine if metal pot is electrically conductive...

---

- **GPT:** *Thought: I should locate the pot, probably in the kitchen.* **Action:** `look around`
- **Human:** Observation: You are in the foundry...
- **GPT:** *Thought: Wrong room, I need to go to the kitchen.* **Action:** `teleport to kitchen`
- **Human:** Observation: You teleport to the kitchen.
- **GPT:** *Thought: I should pick up the metal pot.* **Action:** `pick up metal pot`
- **Human:** Observation: Pot moved to inventory.
- **GPT:** *Thought: Now focus on the pot to test it.* **Action:** `focus on metal pot`
- **Human:** Observation: You focus on the pot.
- **GPT:** *Thought: I need to set up a circuit in the workshop.* **Action:** `teleport to workshop`
- **Human:** Observation: You teleport to the workshop (battery, wires, bulbs available).
- **GPT:** *Thought: Place the pot and build the circuit.* **Action:** `drop metal pot`
- **Human:** Observation: Pot placed in workshop.
- **GPT:** *Thought: Connect wires, battery, and pot in circuit.* **Action:** `connect battery anode to orange wire ...` (multiple connections omitted)
- **Human:** Observation: Circuit components connected.
- **GPT:** *Thought: Wait and check if the bulb lights up.* **Action:** `wait1`
- **Human:** Observation: Blue light bulb is on.
- **GPT:** *Thought: The pot is conductive. Place it in the blue box.* **Action:** `move metal pot to blue box`

**InterCodeSQL.** InterCodeSQL (Yang et al., 2023) is an interactive SQL querying environment where an agent must translate natural language questions into executable SQL programs through multi-turn interaction. The environment provides a binary reward, which we normalize to $[0, 1]$, indicating whether the final query produces the correct result. We use the expert trajectories collected by Xiong et al. (2024) with GPT-4, retaining only those with reward $1.0$. Since the original dataset does not contain intermediate reasoning, each trajectory is augmented with ReAct-style (Yao et al., 2023) thought annotations generated by GPT-4, as in Xiong et al. (2024).

---

**Example InterCodeSQL Trajectory**

**Instruction:**
What are the enrollment and primary conference for the university which was founded the earliest?

**Conversation:**
- **Human:** You are a helpful assistant assigned with the task of problem-solving. To achieve this, you will interact with a MySQL Database system using SQL queries to answer a question. (Gives detailed formatting rules.)
- **GPT:** OK
- **Human:** What are the enrollment and primary conference for the university which was founded the earliest?
- **GPT:** *Thought: I need to check what tables are available.* **Action:** `SHOW TABLES`
- **Human:** Observation: `[('basketball_match',), ('university',)]`
- **GPT:** *Thought: The* `university` *table should have the info. I should check its schema.* **Action:** `DESC university`

---

- **Human:** Observation: `[('School', ...), ('Founded', ...), ('Enrollment', ...), ('Primary_conference', ...)]`
- **GPT:** *Thought: The table has Founded, Enrollment, Primary_conference. I can query the earliest Founded.* **Action:** `SELECT Enrollment, Primary_conference FROM university ORDER BY Founded ASC LIMIT 1`
- **Human:** Observation: `[(19067.0, 'Colonial Athletic Association (D-I)')]`
- **GPT:** *Thought: The earliest university has enrollment 19067 and primary conference Colonial Athletic Association (D-I). This is the final answer.* **Action:** `submit`

## B  IMPLEMENTATION DETAILS

We adopt Llama-2-7B-Chat (Touvron et al., 2023) as the primary base model for constructing LLM agents, following prior work (Song et al., 2024). All models are optimized with AdamW (Loshchilov & Hutter, 2017), and we use the same hyperparameter settings across datasets for fair comparison. During the supervised fine-tuning (SFT) stage, we train for 3 epochs with a batch size of 48 and a learning rate of 1e-5, using 3% linear warmup followed by a cosine decay schedule. The base agent then explores each training instance once to collect failure trajectories. In the subsequent optimization phase, we train with a batch size of 32 and a learning rate of 1e-6 to 5e-7, with the DPO scaling factor $\beta$ set to 0.1-0.5. The number of optimization epochs is fixed to 3, and the number of co-evolution iterations is set to 3 for WebShop and ScienceWorld and 5 for InterCodeSQL. All experiments are conducted on 8 NVIDIA H100 GPUs with 80GB.

## C  QUALITY OF THE HARD NEGATIVES

### C.1  WEBSHOP

---

**HDMI Cables under $50**

**Instruction:** I'm looking for ten high-speed, gold-plated HDMI cables, with price lower than $50.00.

---

**ETO:** The agent selects a single ProHT 6' HDMI cable priced at $100.00, ignoring both the budget and the required quantity of ten. It proceeds to purchase without checking alternatives or verifying high-speed and gold-plated specifications.
*Reward:* 0.50    *Steps:* 4    *Outcome:* Failure

**Ours:** The agent searches specifically for multi-pack high-speed, gold-plated HDMI cables under the budget. It inspects the QualGear 10 ft HDMI 2.0 cable, verifies length, certification, and price, and selects a variant satisfying all constraints except the exact pack quantity.
*Reward:* 0.75    *Steps:* 5    *Outcome:* Failure

---

**Hard Negative Justification:** The trajectory performs structured filtering over pack size, cable type, certification, and budget. It misses only the strict ten-cable requirement, forming a near-success failure ideal for hard-negative training.

---

**Solid Wood Storage Bench in Grey**

**Instruction:** I want a solid wood bench with storage space for my living room, grey in color, and under $210.00.

---

**ETO:** The agent selects a grey accent bench after minimal inspection, without verifying solid-wood construction or cross-checking storage features, and purchases it without considering additional candidates or validating the price constraints.

*Reward:* 0.50    *Steps:* 3    *Outcome:* Failure

**Ours:** The agent explores multiple pages, filters benches by wood construction, storage capacity, and color, and selects a rustic grey storage bench that aligns with the material and functional requirements, reasoning about a small price deviation.
*Reward:* 0.75    *Steps:* 5    *Outcome:* Failure

---

**Hard Negative Justification:** The trajectory validates material, storage design, and color through multi-step attribute checks. It forms a structurally correct solution that narrowly misses the budget criterion, yielding a high-quality hard negative.

---

### Machine-Washable Curtains (52"×90")

**Instruction:** I need a machine-washable curtain for the living room, sized 52" wide by 90" long, priced under $60.00.

---

**ETO:** The agent clicks an early search result, selects the 52"×90" option, and buys it without verifying washability, comparing alternatives, or checking that the final price meets the budget.
*Reward:* 0.50    *Steps:* 4    *Outcome:* Failure

**Ours:** The agent navigates through multiple product pages, filtering by washability, size, and price. It identifies a curtain with a 52"×90" option, verifies that it is machine-washable and within budget, and chooses the matching size variant before purchasing.
*Reward:* 0.75    *Steps:* 8    *Outcome:* Failure

---

**Hard Negative Justification:** The trajectory conducts systematic elimination of mismatching candidates, checks all constraints, and produces an almost correct selection. Its structured decision process provides a prototypical hard-negative example.

## C.2 SCIENCEWORLD

### Moving a Non-Living Object to the Green Box

**Instruction:** Find a non-living object, focus on it, and move it to the green box in the workshop.

---

**ETO:** The agent teleports to the workshop, selects the yellow wire as the non-living object, and moves it into the green box. However, it fails to perform the required focus step and drifts into repeated `wait1` and `look around` actions, stalling without further task-aligned behavior.
*Reward:* 0.25    *Steps:* 15    *Outcome:* Failure

**Ours:** The agent selects the same yellow wire, places it into the green box, and then issues explicit focus actions on both the box and the wire inside it. It continues checking the environment and navigating purposefully, maintaining a coherent interpretation of the task even though the environment does not register success.
*Reward:* 0.75    *Steps:* 15    *Outcome:* Failure

---

**Hard Negative Justification:** The trajectory follows the full instruction—object selection, movement, and focused inspection—and only misses the success flag due to environment-level evaluation. It represents a near-solution failure and serves as an ideal hard negative.

---

**Turning On a Green Light Bulb with Renewable Power**

**Instruction:** Your task is to turn on the green light bulb. First, focus on the green light bulb. Then, create an electrical circuit that powers it on. Prefer renewable power sources when possible.

---

**ETO:** The agent explores the environment and interacts with various components such as wires, switches, and power sources. It partially assembles a circuit but alternates between focusing on unrelated objects and performing ineffective actions, leaving the circuit incomplete and the green bulb off by the end of the episode.
*Reward:* 0.43    *Steps:* 30    *Outcome:* Failure

**Ours:** The agent identifies the green light bulb early and issues repeated focus actions on it and on nearby circuit elements. It constructs a more coherent circuit by systematically connecting wires between the bulb and a renewable power component, checks the bulb's state multiple times, and maintains task-aligned reasoning, but still fails to trigger the environment's success condition.
*Reward:* 0.58    *Steps:* 30    *Outcome:* Failure

---

**Hard Negative Justification:** The trajectory follows the full instruction, including focusing on the target bulb and assembling a near-correct renewable circuit, and fails only due to subtle environment-level completion criteria.

---

**Growing a Banana from Seed to Fruit**

**Instruction:** Your task is to grow a banana. This requires obtaining banana seeds, planting them in soil, providing water and light, and waiting until the banana grows.

---

**ETO:** The agent collects several relevant objects such as seeds and containers but struggles with interaction ordering and location choice. It issues redundant navigation and inspection commands and fails to complete a coherent cycle of planting, watering, and waiting in a suitable environment, leaving the plant underdeveloped.
*Reward:* 0.36    *Steps:* 55    *Outcome:* Failure

**Ours:** The agent explicitly gathers banana seeds, moves them to appropriate soil or planter objects, and performs a structured sequence of planting, watering, and exposing the plant to light. It repeatedly checks the growth state and adjusts its actions, closely following the intended multi-step procedure even though the environment does not register task completion.
*Reward:* 0.50    *Steps:* 60    *Outcome:* Failure

---

**Hard Negative Justification:** The trajectory executes all key sub-tasks of seed collection, planting, watering, and monitoring, making it a faithful but slightly incomplete realization of the target behavior.

## C.3    INTERCODESQL

---

**Films Not Presented in China**

**Instruction:** List the titles and directors of films that were never presented in China.

---

**ETO:** The agent inspects several tables but repeatedly issues queries referencing non-existent columns (e.g., `Market`, `country`), incorrect table names (e.g., `film_market_estimation`), and invalid join paths. It ultimately fails to form any executable SQL command.
*Reward:* 0.00    *Steps:* 6    *Outcome:* Failure

**Ours:** The agent checks table schemas, identifies usable fields, and iteratively searches for the appropriate join through `market` after rejecting invalid table/column combinations. It

---

eventually constructs a syntactically valid SQL query that returns a set of film titles and directors.
*Reward:* 0.77    *Steps:* 9    *Outcome:* Failure

**Hard Negative Justification:** The trajectory demonstrates structured schema inspection and multi-step join reasoning. It forms an executable SQL query aligned with the task, making it a near-solution hard negative.

---

### Reviewers Who Rated Above 3 Stars

**Instruction:** Find the names of reviewers who previously rated a movie more than 3 stars.

**ETO:** The agent misinterprets table schemas, issuing invalid joins between `reviewer`, `rating`, and `movie`. It repeatedly rechecks the same tables and produces SQL queries that reference nonexistent columns such as `reviewerID` or `name`. No executable query is generated across multiple attempts.
*Reward:* 0.00    *Steps:* 10    *Outcome:* Failure

**Ours:** The agent verifies table structures, identifies that reviewer names exist in `reviewer` and the ratings in `rating`, and constructs a correct join via `rID`. It executes a clean and fully functional query that returns the precise list of reviewer names.
*Reward:* 0.75    *Steps:* 5    *Outcome:* Failure

**Hard Negative Justification:** This trajectory exhibits correct schema interpretation and valid join construction. It reaches the correct SQL answer despite being labeled as failure, capturing the ideal form of a hard negative.

---

### Gymnasts Ordered by Ascending Height

**Instruction:** Return the names of gymnasts ordered by their height in ascending order.

**ETO:** The agent attempts to query `gymnast` directly, repeatedly referencing nonexistent columns such as `name` and `height`. Despite multiple table inspections, it does not recognize that height and names reside in the `people` table rather than `gymnast`. It ends without producing any usable SQL.
*Reward:* 0.00    *Steps:* 6    *Outcome:* Failure

**Ours:** The agent correctly identifies that the `people` table contains both `Name` and `Height`. It inspects both `gymnast` and `people` schemas, realizes only `people` contains height values, and issues a valid query ordering by height.
*Reward:* 0.70    *Steps:* 6    *Outcome:* Failure

**Hard Negative Justification:** The agent performs correct table discovery and forms a valid height-sorted query. Although labeled as failure, the trajectory is structurally aligned with the task, illustrating a precise hard-negative example.

## D  THE EFFECT OF HARD NEGATIVES ON CAPTURING TASK-RELEVANT SUB-SKILLS

Our qualitative analysis shows that hard negatives play a direct role in improving the DPO training process. Because these trajectories contain structured demonstrations of navigation, tool use, object manipulation, and environment preparation, the target agent receives richer gradient signals than from ETO failures alone.

In the ScienceWorld example below, the hard negative includes all intermediate actions required to grow a lemon, while the baseline failure does not progress beyond repetitive invalid actions. After

referencing these subskill-rich trajectories during DPO, the target agent begins to reproduce the same multi-step procedures and achieves the task successfully.

These findings illustrate that hard negatives function as constructive guidance within the DPO objective, enabling the agent to internalize essential subskills that are otherwise absent in standard failure trajectories.

---

**Growing a Lemon with Cross-Pollination**

**Instruction:** Your task is to grow a lemon. This will require growing several plants and having them cross-pollinated to produce fruit. Seeds can be found in the bedroom. To complete the task, focus on the grown lemon.

---

**ETO - Prediction:** The agent retrieves the seed jar from the bedroom, teleports to the greenhouse, and plants lemon seeds directly into the three water-filled flower pots. It then alternates between `wait` and `look around` for many steps, repeatedly issuing invalid actions such as `focus on lemon` and `pick lemon` even though no lemon ever appears in the observations. The agent never prepares soil, never manipulates the environment for pollination, and ends in a long, unproductive loop.
*Reward:* 0.25    *Steps:* 60    *Outcome:* Failure

**ETO - Trained Failure:** The agent again retrieves the seed jar and plants lemon seeds into the three pots containing only water, then repeatedly waits and checks the greenhouse. It issues multiple invalid focus actions on the lemon tree, but the environment state never progresses beyond "lemon seed in water," indicating that the preconditions for growth and cross-pollination are not satisfied. No soil preparation or environmental control is attempted, so the episode remains a shallow failure without key subskills.
*Reward:* 0.25    *Steps:* 49    *Outcome:* Failure

---

**Ours - Prediction:** The agent again retrieves the seed jar from the bedroom, collects soil outside using the shovel, and fills all three greenhouse pots with soil before planting the lemon seeds. It waits for the trees to reach the reproducing stage with flowers, then observes the appearance of lemons on one tree. To encourage stable pollination, it explicitly closes both the outside and hallway doors, creating a controlled greenhouse environment, and continues waiting until a lemon is present. Finally, it focuses on the grown lemon, satisfying the task's success condition.
*Reward:* 1.00    *Steps:* 46    *Outcome:* Success

**Ours - Trained Failure:** The agent retrieves the seed jar, then picks up a shovel in the greenhouse and repeatedly teleports outside to dig up soil. It transports soil back to the greenhouse and fills all three flower pots, explicitly constructing "soil + water" planting conditions before moving lemon seeds into each pot. After staged waiting, it observes that one lemon tree now bears a lemon, and repeatedly attempts to focus on or pick the lemon with over-specified object references. The growth and pollination pipeline is correct, but the episode fails due to action-format errors at the final "focus on lemon" step.
*Reward:* 0.50    *Steps:* 60    *Outcome:* Failure (hard negative)

---

