# OpenReview forum: "Co-Evolving Agents: Learning from Failures as Hard Negatives"
_ICLR.cc/2026/Conference — Submitted to ICLR 2026_

### Official Review · Reviewer_AXJg · 2025-10-26

**Soundness:** 3
**Presentation:** 3
**Contribution:** 2
**Rating:** 4
**Confidence:** 3

**Summary:**

To address overfitting in self-improving agents with scarce data, the authors propose a novel co-evolving agents framework. The target agent is improved jointly with an auxiliary failure agent that uses preference optimization to generate informative hard negatives. Incorporating these hard negatives sharpens decision boundaries, significantly enhancing generalization and performance.

**Strengths:**

1) Well motivated. Due to scarce data, the authors try to improve target agent with a failure agent, that generate hard negative samples.
2) The improvement seems significant. Results achieve 5% average improvements compared to the best baseline.

**Weaknesses:**

1) The current experimental setup appears limited. The evaluation is conducted solely on the Llama-2-7B architecture instead of more recent model architectures (e.g., Llama-3, Qwen-3) and larger model scales.

2) The current ablation study lacks a critical control experiment. The proposed co-evolving framework inherently doubles the number of trainable parameters (Target Agent + Failure Agent) compared to a standard single-agent baseline (e.g., a 7B model). The authors must conduct an ablation to isolate whether the observed performance gains are due to the co-evolutionary mechanism itself or simply the increased parameter count. Specifically, an experiment comparing the following two conditions is necessary:
* 7 B Target Agent + 7 B Failure Agent.
* A single 14B model trained under the standard SFT or RFT.

**Questions:**

See weaknesses.

---

> ### Author Response · Authors · 2025-11-27
> **Response to Reviewer AXJg**
>
> We thank the reviewer for the valuable and constructive feedback. The comments have substantially strengthened the paper and improved its clarity. We've updated the paper to thoroughly address all concerns, with the corresponding changes highlighted in blue.
>
> ---
>
> ### [W1] More recent model architectures (e.g., Llama-3, Qwen-3) and larger model scales.
>
> Thank you for the comment. To address this concern, we extended our evaluation to a more recent model architecture. As shown in the table below, using Qwen3-4B-Instruct-2507, our method consistently outperforms ETO across all benchmarks, improving WebShop, ScienceWorld (seen and unseen), and InterCodeSQL by 6.8%, 6.5%, 3.3%, and 10.3%, respectively. These results demonstrate that our framework remains effective beyond Llama-2-7B and generalizes well across architectures. We’ve included this results in Sec. 5.3 (Line 466-479).
>
> | Models                           | WebShop | ScienceWorld (Seen) | ScienceWorld (Unseen) | InterCodeSQL | Avg.  |
> |-----------------------------------|---------|----------------------|------------------------|--------------|-------|
> | Qwen3-4B-Instruct-2507 + SFT      | 63.9    | 43.6                 | 40.8                   | 12.7         | 40.3  |
> | Qwen3-4B-Instruct-2507 + ETO      | 65.7    | 58.6                 | 55.2                   | 58.8         | 59.5  |
> | **Qwen3-4B-Instruct-2507 + Ours** | **72.5**| **65.1**             | **58.5**               | **69.1**     | **66.3** |
>
> We’re also running experiments on Llama-2-13B to examine larger model scales and will update the results in the discussion period once they are complete.
>
> ---
>
> ### [W2] A single 14B model trained under the standard SFT or RFT.
>
> Thank you for the thoughtful suggestion. First, we would like to clarify that our original submission included a parameter-matched ablation where the failure agent is replaced with a 7B auxiliary positive agent trained in the same manner as ETO ('7B positive agent + 7B positive agent'). In this setting (“7B target agent + 7B positive agent’’), the model achieves an average reward of 62.76 on WebShop, which is nearly identical to ETO and 3.9% lower than our method(“7B target agent + 7B failure agent’’). Additionally, the Llama-2-13B under the SFT achieves 59.3 on WebShop, which is still below our 7B+7B failure-agent result 66.7.
>
> Given that pretrained LLM prior knowledge is a major factor in performance, and considering that our ‘7B positive-agent + 7B positive-agent’ experiment shows almost no improvement, these results show that the performance gains stem from the informative hard negatives produced by the failure agent rather than from simply adding more parameters. To avoid confusion, we've clarified this point more explicitly in Section 5.4 (Line 519–526) of the revision.
>
> As mentioned in the response to [w1], we're currently running our method on Llama-2-13B as well, and we’ll share the results as soon as they become available.
>
> ---

---

> > ### Comment · Reviewer_AXJg · 2025-11-28
> >
> > I have no further questions. The newly added experiments on Qwen3-4B-Instruct-2507 solved my concerns.
> >
> > However, the rating can not be changed now.

---

> > > ### Author Response · Authors · 2025-11-28
> > >
> > > We’re glad to hear that our response addressed your concerns.
> > > We understand that the rating cannot be changed due to the updated policy. Thank you again for your thoughtful feedback and for taking the time to review our work!

---

> ### Author Response · Authors · 2025-11-28
>
> Additionally, we evaluated our method on the larger Llama-2-13B model and observed consistent improvements. As shown in the table below, our method outperforms both SFT and ETO by a substantial margin on WebShop and ScienceWorld, confirming that the performance gains persist even at larger model scales. We'll include the related experiments in the revised manuscript.
>
> | Models                   | WebShop | ScienceWorld (Seen) | ScienceWorld (Unseen) |
> |--------------------------|---------|----------------------|------------------------|
> | Llama-2-13B-chat + SFT   | 59.3    | 64.49               | 56.06                 |
> | Llama-2-13B-chat + ETO   | 65.0    | 72.61               | 65.27                 |
> | Llama-2-13B-chat + Ours  | **69.2** | **74.45**           | **65.05**             |

---

### Official Review · Reviewer_EjWj · 2025-10-27

**Soundness:** 2
**Presentation:** 2
**Contribution:** 2
**Rating:** 4
**Confidence:** 3

**Summary:**

This paper introduces a co-evolving agents framework designed to enhance the learning of self-improving language model agents by leveraging failure trajectories as informative training signals. The key idea is to pair a target agent with a failure agent that learns to generate hard negatives—failed trajectories that are close to success. Unlike existing self-improving frameworks such as Exploration-Based Trajectory Optimization (ETO), which risk overfitting due to limited ground-truth supervision, the failure agent here continuously learns from both its own and the target’s failure trajectories through Direct Preference Optimization (DPO). These refined hard negatives are incorporated into the target agent’s preference optimization, sharpening decision boundaries and improving generalization. The authors demonstrate their method on the WebShop, ScienceWorld, and InterCodeSQL benchmarks. Their results show improved performance over baselines like ETO, particularly a significant +10.8 point gain in generalization on unseen ScienceWorld tasks.

**Strengths:**

1. The idea of a failure agent that co-evolves with the main model is novel and well-motivated.

2. The theoretical setup (POMDP formalization, DPO-based optimization) is rigorous, and the experimental evidence convincingly supports the claims.

3. The approach directly addresses a key limitation of current self-improving LLM agents—overfitting to limited expert data—by leveraging a sustainable, self-generated source of supervision. Results across multiple domains, along with both quantitative and qualitative failure analysis, strengthen credibility.

**Weaknesses:**

1. The method relies on a multiple-stage process (SFT → failure-agent DPO on failure–failure pairs → target-agent DPO+SFT on mixed pairs) executed in alternating iterations, but Figure 1 does not illustrate this flow or the data construction steps, making Section 4 harder to follow at a glance.

2. The most significant weakness is in Section 5.2.1. The text states it will provide a qualitative comparison of failure trajectories, describing the ETO baseline's "degenerate failure" and contrasting it with the "Ours" method's "more structured failure trajectories". However, the example provided for "Ours" is explicitly labeled as a "Success" (Reward: 0.75, Outcome: Success). This example fails to illustrate what a "hard negative failure" from the proposed method actually looks like and directly contradicts the text's stated purpose.

3. The InterCodeSQL results show no improvement, which the authors attribute to sparse rewards. This suggests the method's applicability may be limited to environments with dense reward signals.

4. Appendix Section C seems incomlete.

**Questions:**

1. Training two agents alternately is presumably more expensive than training one. How much does this increase training time and compute requirements? Is the performance gain worth the added cost?

2. What prevents the failure agent from collapsing to a non-productive policy? For example, what stops it from becoming identical to the target agent and only learning to succeed, or conversely, learning a trivial policy that always fails with a reward of 0? How does it maintain the "near-success" decision boundries while enabling generating a larger pool of failures?

3. Could you provide more detail on the training dynamics in Section 4.3? Specifically, when constructing the target agent's dataset $D_{
tgt}$, how are the three types of preference pairs—(expert, target-failure), (expert, failure-agent-failure), and (failure-failure pairs)—sampled and balanced during training?

---

> ### Author Response · Authors · 2025-11-27
> **Response to Reviewer EjWj (1 / 2)**
>
> We thank the reviewer for the insightful and helpful feedback. The comments have substantially strengthened the paper and improved its clarity. We've updated the paper to thoroughly address all concerns, with the corresponding changes highlighted in blue.
>
> ---
>
> ### [Q1] How much does this increase training time and compute requirements?
>
> Since the target and failure agents are trained in alternating steps, the compute cost is about twice that of training a single agent. However, given the recent momentum in multi-agent self-improving frameworks, we view this level of compute scaling as reasonable for the corresponding performance gains, and assigning one agent the role of a failure generator within a multi-agent setting represents a reasonable research direction.
>
> ---
>
> ### [Q2] What prevents the failure agent from collapsing to a non-productive policy?
>
> Although the failure agent is trained on failure trajectories, we observed that the failure agent doesn't collapse. This is because it is jointly trained on the target agent’s failure trajectories, which still reflect meaningful attempts toward the ground-truth success trajectories and therefore provide meaningful learning signals. These partially successful behaviors anchor the failure agent toward competent action sequences as weak supervision and prevent it from drifting into trivial or repetitive failures. We've clarified this point in Section 5.2.1 (Line 380–384) in revision.
>
> ---
>
> ### [Q3] More details on the training dynamics in Section 4.3
>
> We apologize for the confusion. First, following prior works, we perform only one rollout per instruction due to the high cost and long multi-turn trajectories. Consequently, the total number of generated trajectories equals the number of expert trajectories in the dataset.
>
> For clearer understanding, we've provided the overall statistics of expert trajectories (Table 1) together with an analysis comparing the ratios of successful, negative, and hard-negative trajectories generated by the agents during training. This allows us to examine how co-evolution affects the balance among success, near-success (hard negatives), and failure trajectories, and how this balance relates to the observed performance improvements.
>
> Across all benchmarks, our method generates substantially more negative and hard-negative trajectories than ETO. Specifically, negative trajectories increase by 9.5% in WebShop, 16.7% in ScienceWorld, and 9.0% in InterCodeSQL, while hard negatives increase by 2.3%, 8.7%, and 4.3%, respectively. This shift in data balance effectively expands the informative failure space, providing stronger preference signals and contributing to the performance gains of our method. We've added these results in Table 1&2 and Section 5.2.1 (Line 353–361).
>
> Table 1. Overview of the benchmark statistics.
> | Dataset       | Train | Test-Seen | Test-Unseen | Action Space   | Max Turns |
> |--------------|-------|-----------|-------------|----------------|-----------|
> | WebShop      | 1624  | 200       | --          | 8              | 10        |
> | ScienceWorld | 1483  | 194       | 241         | 16             | 100       |
> | InterCodeSQL | 1500  | 200       | --          | ∞ (SQL)        | 10        |
>
> Table 2. Statistics of generated trajectories.
> | **Task**        | **Method** | **Success ↓** | **Failure ↑** | **Hard Negative ↑** |
> |-----------------|------------|---------------|----------------|----------------------|
> | **WebShop**     | ETO        | 51.4%         | 25.9%          | 22.7%               |
> |                 | Ours       | **39.6%**     | **35.4%**      | **25.0%**           |
> | **ScienceWorld**| ETO        | 75.32%        | 19.29%         | 5.39%               |
> |                 | Ours       | **49.90%**    | **36.01%**     | **14.09%**          |
> | **InterCodeSQL**| ETO        | 58.67%        | 37.73%         | 3.20%               |
> |                 | Ours       | **45.80%**    | **46.73%**     | **7.47%**           |
>
> ---
>
> ### [W1] The main figure should illustrate the multi-stage process.
>
> Thanks for pointing this out. We've revised the main figure (Fig. 1) to clearly represent the multi-stage pipeline and the associated data construction steps, making Sec. 4 easier to follow.
>
> ---
>
> ### [W2] a qualitative comparison of failure trajectories
> Apologize for the confusion. Labeling “Ours” as “Success” was a typo. In the revision, we’ve updated and added qualitative comparisons between failures and hard negatives in Sec.5.2.2 (Line 390-412) and Appendix C. These examples show that our hard negatives contain substantially more instruction-relevant behaviors than the failures produced by ETO.
>
> ---

---

> ### Author Response · Authors · 2025-11-27
> **Response to Reviewer EjWj (2 / 2)**
>
> ### [W3] The InterCodeSQL results show no improvement
>
> Thank you for raising this issue. The lack of improvement on InterCodeSQL has been addressed. In the initial version of our experiments, length-slicing for failure trajectories was not included, even though this procedure is commonly applied in the baselines including ETO. After applying the same length-slicing strategy, the training became much more stable, and we observed a 4.4% improvement over ETO on InterCodeSQL (in Table 3).
> (More specifically, when non-preferred trajectories are substantially longer than preferred ones, DPO can unintentionally favor shorter sequences rather than actual content, leading to instability. Prior work mitigates this by slicing non-preferred trajectories so that their lengths do not deviate excessively from the preferred ones, and we now follow the same practice.)
>
> In addition, as noted earlier, in sparse-reward settings our method provides relatively smaller benefit compared to dense-reward settings. Since our framework relies on distinguishing hard negatives based on reward feedback, tasks where rewards cannot meaningfully differentiate failure quality inherently offer less signal. Because this challenge stems from the reward structure itself, approaches that provide denser feedback could naturally complement our method. For instance, using step-level reward estimation as in IPR to form a denser reward setting may provide a useful avenue for improving generalization under sparse-reward environments.
>
> ---
>
> ### [W4] Appendix
> Thanks for pointing this out. We have updated the appendix to provide clearer dataset specifications and additional qualitative results.

---

### Official Review · Reviewer_enAe · 2025-11-01

**Soundness:** 2
**Presentation:** 3
**Contribution:** 2
**Rating:** 4
**Confidence:** 4

**Summary:**

The paper proposes a co-evolving framework with two policies: a target agent trained via preference optimization and a specialized failure agent trained on failure-vs-failure preference pairs to synthesize hard negatives (near-success but still failing trajectories). By alternating updates, the failure agent continually produces informative negatives that sharpen the target agent’s decision boundary. Experiments on WebShop, ScienceWorld, and InterCodeSQL report consistent gains over common preference/RL baselines (notably on unseen splits), with qualitative analyses suggesting the method increases the density and quality of “useful failures.”

**Strengths:**

Clear, intuitive idea: Elevating “failures” into structured supervision via failure-vs-failure preferences is conceptually neat and practically motivated.

Co-evolution design: Alternating updates between a failure generator and a target learner is a simple mechanism to keep training signals challenging and fresh.

Analyses beyond toplines: The paper examines failure quality/quantity and includes ablations (e.g., replacing the failure agent with a standard “positive” agent), which supports the central claim that specializing on failures matters.

**Weaknesses:**

Limited Novelty in Core Idea: The framework builds heavily on existing methods like ETO (Exploration-Based Trajectory Optimization) and DPO, primarily adding a "failure agent" for hard negatives. While this is an incremental improvement, it may not be sufficiently novel for ICLR, as similar concepts (e.g., negative agents in multi-agent systems or hard negatives in contrastive learning) are referenced in related work but not deeply differentiated.

Baselines are not strong enough as configured.

Prior work shows that RFT (or related preference-optimization variants) can be substantially stronger when you multi-sample rollouts and apply DART-style data selection (e.g., sample several candidates per prompt/step and keep the most informative pairs). In practice, “RFT + multi-sample + DART selection” can outperform ETO; using only ETO (or lightly tuned RFT) as the strongest baseline likely understates what a robust, compute-matched preference pipeline can do.

The paper cites Iterative step-level Process Refinement (IPR), but does not provide a compute-matched, carefully tuned head-to-head. Given IPR’s strong step-level refinement on agentic tasks, a direct comparison (same models, prompts, sampling budget, and filtering strategy) is necessary to substantiate superiority.

Computational Overhead: Training two co-evolving agents alternately requires significant resources (e.g., 8 NVIDIA H100 GPUs for experiments).

Overfitting Risks in Failure Agent: The failure agent is trained solely on failure trajectories, which could lead to overfitting to specific failure modes rather than exploring a broad "failure landscape." The paper claims it generates hard negatives, but empirical evidence is limited to one qualitative example and basic quantitative stats (e.g., Figure 2), without deeper analysis like diversity metrics or ablation on failure pair construction.

Imbalanced Training Signals: The target agent's loss combines DPO and SFT with weights (e.g., λDPO=0.5, λSFT=0.5 for expert pairs), but justification for these hyperparameters is weak. As noted (citing Yuan et al., 2025a), DPO can be unstable due to imbalance between chosen and rejected trajectories, yet no sensitivity analysis or alternatives (e.g., other preference methods like GRPO) are explored.

**Questions:**

Novelty and Differentiation from Prior Work: The failure agent concept builds on ideas like hard negatives from contrastive learning (e.g., Robinson et al., 2021) and negative agents in multi-agent systems (e.g., Zhang et al., 2024). What are the key theoretical or empirical differences that make your co-evolutionary approach distinct from these, beyond the alternating training phases? For instance, does it provably converge or provide guarantees on generating harder negatives over iterations?

Failure Agent Behavior: The qualitative example in Section 5.2.1 shows the failure agent generating more structured near-success failures. However, the quantitative analysis (Figure 2) is limited to trajectory counts and average rewards. Have you evaluated diversity metrics (e.g., trajectory entropy or edit distance) to confirm that the failure agent explores a broad failure landscape rather than overfitting to specific modes? What prevents the failure agent from collapsing into trivial or repetitive failures during co-evolution?

---

> ### Author Response · Authors · 2025-11-27
> **Response to Reviewer enAe (1 / 2)**
>
> We thank the reviewer for the constructive and thoughtful feedback. The comments have substantially strengthened the paper and improved its clarity. We've updated the paper to thoroughly address all concerns, with the corresponding changes highlighted in blue.
>
> ---
>
> ### [Q1 & W1] Novelty of the idea
>
> Thank you for the insightful comment. While our method builds on components such as ETO (self-improving agents that leverage their own failures) and the notion of hard negatives (conventional contrastive learning), the key novelty lies in enabling the agent to autonomously generate informative hard negatives without external supervision, even in a setting where it must self-improve with only a very limited number of expert trajectories across complex and long multi-turn tasks (web shopping, scientific reasoning, SQL).
>
> A self-improving agent ideally requires no human intervention; however, for challenging downstream tasks where pretrained LLMs perform poorly, collecting high-quality trajectories for retraining is typically a major bottleneck. Our contribution is to show that an agent can generate and leverage informative, high-quality failure trajectories, rather than relying solely on the scarce success trajectories or trivial failures, without any human or external supervision. We’ve clarified this in the revision in Line 83-85 and 88-95.
>
> ---
>
> ### [Q2 & W5] Quantitative analysis with diversity metrics & The risk of collapsing into trivial failures
>
> Regarding collapse, we observed that the failure agent doesn't collapse. This is because it is jointly trained on the target agent’s failure trajectories, which still reflect meaningful attempts toward the ground-truth success trajectories and therefore provide meaningful learning signals. These partially successful behaviors anchor the failure agent toward competent action sequences as weak supervision and prevent it from drifting into trivial or repetitive failures. We've clarified this point in Section 5.2.1 (Line 380–384) in revision.
>
> Additionally, in response to the reviewer’s insightful feedback, we conducted  a quantitative diversity analysis and added in the revision (Line 373-380). We embed each generated failure trajectory using the LLM2Vec–Meta-Llama-3-8B-Instruct-mntp as a sentence-embedding model and compute the average pairwise distance among trajectories generated per instruction during training.
> As shown in Figure 3 in the revision, the failure agent produces a larger number of trajectories with higher distances compared to ETO, and the boxplot demonstrates a notably larger mean and standard deviation. This confirms that the failure agent explores a broader and more diverse failure space rather than overfitting to specific modes. In the table below, we report the mean and standard deviation of these pairwise distances, showing that our method indeed covers a wider failure landscape.
>
> | Method | Avg Distance | Std Dev |
> |--------|--------------|---------|
> | ETO    | 0.0133       | 0.0161  |
> | Ours   | **0.0202**       | **0.0184**  |
>
> ---
>
> ### [W2] Additional baseline: RFT+multi-sample+DART selection.
>
> Thank you for the constructive suggestion. Following the reviewer’s recommendation, we conducted additional experiments with RFT combined with multi-sample rollouts and DART-style data selection. Because our target tasks involve long multi-turn trajectories, we set the RFT multi-sample budget to N=10 and used n=5 for computing the difficulty in DART, which already increases the rollout cost by about 10-15X. On WebShop, the results are as follows:
> - RFT: 60.89
> - RFT + DART: 61.90
>
> We observe a modest improvement from applying DART, but the gains are not as strong as those reported in math-oriented settings. We believe this difference arises from the small sampling budgets or from the nature of the tasks: DART is particularly effective when pretrained LLMs already possess substantial prior knowledge (e.g., arithmetic and algebra), making it more likely that multi-sampling produces some correct answers. In contrast, our tasks including web shopping, scientific reasoning, and interactive SQL require environment-specific reasoning that goes beyond the prior knowledge encoded in pretrained LLMs, reducing the likelihood that multi-sampling yields success trajectories. We’ve added this discussion to the revision in Sec. 5.3 (Line 481-511).

---

> ### Author Response · Authors · 2025-11-27
> **Response to Reviewer enAe (2 / 2)**
>
> ---
>
> ### [W3] Experiments on Iterative step-level Process Refinement (IPR)
>
> Thank you for the suggestion. IPR is an extension of ETO that improves performance through step-level reward estimation, and it is orthogonal to our approach, which focuses on generating informative failure trajectories. However, given its potential complementarity, we are currently integrating IPR into our pipeline. Since IPR requires multiple rollouts per instruction, the computation cost is substantial, but we'll report the results as soon as possible if they are ready within the discussion period.
>
> ---
>
> ### [W4] Computational Overhead
>
> Previous works such as ETO and IPR also operate under the same 8×A100/H100 setting, as our target tasks involve long, multi-step trajectories and require applying DPO over these sequences, which necessitates high-memory GPUs. In terms of computation time, our method introduces additional cost due to the extra agent, but this overhead results in clear performance gains and points to a promising direction for scaling multi-agent self-improving research.
>
> ---
>
> ### [W6] Sensitivity Analysis on Loss Weighting
>
> We follow the SFT–DPO weighting scheme used in prior work such as IPR. In response to the reviewer’s comment, we conducted a sensitivity analysis by fixing the DPO weight at 1.0 and varying the SFT weight across $\lambda_{\text{SFT}} \in {0.01, 0.1, 0.5}$. As shown in the table below, our method remains reasonably stable as long as the SFT term is kept sufficiently weaker than DPO. This analysis has been added to Sec. 5.4 (Line 515–518).
>
> | λ_SFT | Avg. Reward |
> |-------|-------------|
> | 0.01  | 66.98   |
> | 0.1   | 66.70       |
> | 0.5   | 66.50       |

---

### Official Review · Reviewer_SFFh · 2025-11-02

**Soundness:** 3
**Presentation:** 3
**Contribution:** 3
**Rating:** 4
**Confidence:** 4

**Summary:**

This paper proposes a co-evolution framework with two agents: a target agent that is ultimately optimized for performance and a dedicated failure agent that learns a fine-grained model of the failure landscape. Both agents are initialized via behavior cloning, then independently tuned with SFT before co-evolution. The failure agent is trained with DPO on preference data obtained from failure trajectories generated by itself and the target agent. Once the failure agent reliably produces hard negatives, the target agent is updated using a weighted DPO objective on those negatives alongside SFT on expert-prediction and failure-failure pairs, plus an auxiliary SG loss to stabilize learning and anchor high-reward behaviors. The core idea is to use noisy interaction data as structured supervision that sharpens decision boundaries, and the results demonstrate improvements over baselines.

**Strengths:**

1.	The paper is clearly written and provides a strong motivation for converting interaction data into structured supervision using failure trajectories to learn strong decision boundaries.  The co-evolution learning process using hard negatives that are near-success failures promotes sharper decision boundaries and finer discrimination.
2.	The paper includes an extensive evaluation across different tasks. The comparisons with baseline methods demonstrate the impressive performance of the proposed framework. The ablation studies further highlight the importance of each module in the method.
3.	The problem formulation using co-evolution learning process by utilizing hard negatives is an interesting idea, and this could serve as foundation for future research.

**Weaknesses:**

1.	Hard negatives: It is unclear what hard negative quantitatively mean. They are defined as trajectories that are closer to success but still unsuccessful. However, it’s unclear how close to success is quantitatively defined, is it based on reward threshold?
2.	One of the claims of the paper is that the limited number of expert trajectories result in overfitting. However, no information about the number of successful trajectories versus the hard negatives generated trajectories is provided to understand the statistics and how much does the co-evolution result in creating this balance of close to success, success and failure trajectories. And how these numbers impacts the overall performance? It is unclear how data balance impacts performance and overfitting claims.
4.	Qualitative analysis section mentions that generating hard negatives make the agent capture subskills such as navigation, object manipulation and device control, however, no qualitative results in any of the benchmarks demonstrate that. It is a critical part of the paper, as it shows the importance of using hard negatives for co-evolution.
5.	Further, no qualitative comparisons among expert, hard-negative, and generated/predicted trajectories demonstrate the “near-success” informativeness.
6.	Missing baseline: DPO without co-evolution to isolate the contribution of the failure agent.
7.	Supplementary material appears incomplete; more implementation details and dataset/task specs should be included.

**Questions:**

Please see weaknesses above

---

> ### Author Response · Authors · 2025-11-27
> **Response to Reviewer SFFh (1 / 2)**
>
> We thank the reviewer for the constructive and valuable feedbacks. The comments have substantially strengthened the paper and improved its clarity. We've updated the paper to thoroughly address all concerns, with the corresponding changes highlighted in blue.
>
> ---
>
> ### [W1] Hard negatives: It is unclear what hard negatives quantitatively mean.
> We apologize for the unclear explanation. In our work, we define hardness based on the relative reward value rather than applying a fixed threshold for identifying and leveraging hard negatives. For example, within the 0–1 reward range where success is 1.0, a failure trajectory with a reward 0.95 is treated as harder than one with 0.5 because it lies closer to the success reward 1.0. We've clarified this in the revision (Line 247-250).
>
> ---
>
> ### [W2] The effect of trajectory balance on overfitting is unclear
>
> In response to the reviewer’s constructive feedback, we've provided the overall statistics of expert trajectories (Table 1) together with an analysis comparing the ratios of successful, negative, and hard-negative trajectories generated by the agents during training (Table 2). This allows us to examine how co-evolution affects the balance among success, near-success (hard negatives), and failure trajectories, and how this balance relates to the observed performance improvements.
>
> In this analysis, we separate negative and hard-negative cases using a reward threshold of 0.6 for WebShop and ScienceWorld, and 0 for InterCodeSQL. The reward range is 0 to 1, where successful trajectories receive a reward of 1. Although a higher threshold would be ideal, failures with reward above 0.7–0.8 occur in fewer than 1% of cases in WebShop and ScienceWorld. In InterCodeSQL, the reward structure is highly sparse (most failures receive 0), making higher thresholds infeasible. Following prior works, we perform only one rollout per instruction due to the high cost and long multi-turn trajectories. Consequently, the total number of generated trajectories equals the number of expert trajectories in the dataset.
>
> Across all benchmarks, our method generates substantially more negative and hard-negative trajectories than ETO. Specifically, negative trajectories increase by 9.5% in WebShop, 16.7% in ScienceWorld, and 9.0% in InterCodeSQL, while hard negatives increase by 2.3%, 8.7%, and 4.3%, respectively. This shift in data balance effectively expands the informative failure space, providing stronger preference signals and contributing to the performance gains of our method. We've added these results in Table 2&3 and Section 5.2.1 (Line 353–361).
>
> Table 1. Overview of the benchmark statistics.
> | Dataset       | Train | Test-Seen | Test-Unseen | Action Space   | Max Turns |
> |--------------|-------|-----------|-------------|----------------|-----------|
> | WebShop      | 1624  | 200       | --          | 8              | 10        |
> | ScienceWorld | 1483  | 194       | 241         | 16             | 100       |
> | InterCodeSQL | 1500  | 200       | --          | ∞ (SQL)        | 10        |
>
> Table 2. Statistics of generated trajectories.
> | **Task**        | **Method** | **Success ↓** | **Failure ↑** | **Hard Negative ↑** |
> |-----------------|------------|---------------|----------------|----------------------|
> | **WebShop**     | ETO        | 51.4%         | 25.9%          | 22.7%               |
> |                 | Ours       | **39.6%**     | **35.4%**      | **25.0%**           |
> | **ScienceWorld**| ETO        | 75.32%        | 19.29%         | 5.39%               |
> |                 | Ours       | **49.90%**    | **36.01%**     | **14.09%**          |
> | **InterCodeSQL**| ETO        | 58.67%        | 37.73%         | 3.20%               |
> |                 | Ours       | **45.80%**    | **46.73%**     | **7.47%**           |
>
> ---
>
> ### [W3-4] Lack of qualitative comparisons among trajectories and qualitative results on subskill capturing
>
> Thank you for the constructive suggestion. In the revision, we've added qualitative comparisons between generated failures and hard negatives in Sec.5.2.2 (Line 390-411) and Appendix C. These examples show that our hard negatives contain substantially more instruction-relevant behaviors than the failures produced by ETO. We also include qualitative analyses demonstrating that these hard negatives help the target agent acquire a broader set of sub-skills, as presented in Sec.5.2.2 (Line 413-443) and Appendix D.
>
> ---

---

> ### Author Response · Authors · 2025-11-27
> **Response to Reviewer SFFh (2 / 2)**
>
> ### [W5] DPO without co-evolution to isolate the contribution of the failure agent.
>
> We apologize for the confusion. First, training a single target agent with DPO using only its own failure trajectories corresponds to ETO. In addition, to isolate the contribution of the failure agent while keeping the parameter budget identical, our original submission included an ablation experiment where the failure agent is replaced with an auxiliary positive agent trained in the same manner as ETO (i.e., on expert–prediction pairs only). This setup is effectively DPO without co-evolution but with the same total number of parameters as our method.
>
> This variant achieves an average reward of 62.8 on WebShop, which is nearly identical to ETO and clearly below the 66.7 obtained with our failure-agent framework. These results show that the performance gains stem from the informative hard negatives produced by the failure agent rather than from simply adding more parameters.  To avoid confusion, we've clarified this point more explicitly in Section 5.4 (Line 519–527) of the revision.
>
> ---
>
> ### [W6] Appendix
>
> Thanks for pointing this out. We've updated the appendix to provide clearer dataset specifications and additional qualitative results.

---

### Meta-Review · Area_Chair_ewFj · 2025-12-29

**Summary:**

This work introduces a co-evolving agents framework for the purpose of enhancing the learning of self-improving language model agents that makes use of failure trajectories as informative training signals. The target agent is improved jointly with an auxiliary failure agent using preferences to generate informative hard negative answers. These hard negative answers sharpen the decision boundaries producing better generalization and performance.

The main concerns raised by reviewers are around the limited nature of the experimental setup. An interesting point raised by reviewers is that the proposed framework by its very nature using co-evolving networks actually uses double the parameters than a single model. A crucial experimental omission pointed by the reviewers is the lack of DPO comparison. Other concerns listed were around the novelty of the work.

The experimental ablations added during the rebuttal provided meaningful improvement in the resulting manuscript for example, these solved reviewer AXJg’s concern. Other reviewers did not have a chance to meaningfully interact with the authors around this topic.
This was a difficult decision because from the author’s rebuttal and the reviewers limited discussion it did seem that a substantial improvement in the reviewer’s perception of this work could have occurred, but also weighing on the number of modifications that would be required and the tepid initial response, it is hard to argue for acceptance as is.
I very much encourage the authors to incorporate the feedback gathered during the rebuttal process to make their manuscript better.

**Reviewer Concerns:**

The reviewers raised concerns about the novelty of the work and the experimental results. Some of the experimental concerns were resolved for a subset of the reviewers during the rebuttal, but it is unclear if it would have satisfied all of them.

**Reviewer Scores:**

The score of reviewer AXJg would have certainly been raised to an accept level. It is unclear how would the scores of the other reviewers would have changed due to a lack of interaction between the authors and them. Unfortunately because the initial reviews were all concurrent in their tepid enthusiasm for this work, it is hard to argue they would have meaningfully changed their assessment of the paper.

---

### Decision · Program_Chairs · 2026-01-26

Reject